# Synthesis and Characterization of a Mg^2+^-Selective Probe Based on Benzoyl Hydrazine Derivative and Its Application in Cell Imaging

**DOI:** 10.3390/molecules26092457

**Published:** 2021-04-23

**Authors:** Chunwei Yu, Yuxiang Ji, Shaobai Wen, Jun Zhang

**Affiliations:** 1Laboratory of Environmental Monitoring, School of Tropical and Laboratory Medicine, Hainan Medical University, Haikou 571101, China; cwyu1979@163.com (C.Y.); jiyuxiang@hainmc.edu.cn (Y.J.); wenshaobai@163.com (S.W.); 2Laboratory of Tropical Biomedicine and Biotechnology, Hainan Medical University, Haikou 571101, China

**Keywords:** fluorescent probes, Mg^2+^, cell imaging

## Abstract

A simple benzoyl hydrazine derivative **P** was successfully synthesized and characterized as Mg^2+^-selective fluorescent probe. The binding of **P** with Mg^2+^ caused an obvious fluorescence enhancement at 482 nm. The fluorescent, UV-vis spectra, ^1^H-NMR, and IR spectra confirmed the formation of **P**-Mg^2+^ complex, and the formation of a 1:1 stoichiometry complex was proved by Job’s plot and mass spectrometry. The recognition mechanism of **P** to Mg^2+^ was owing to the photoinduced electron transfer effect (PET). The fluorescent response was linear in the range of 0.9–4.0 µM with the detection limit of 0.3 µM Mg^2+^ in water–ethanol solution (1:9, *v*:*v*, pH10.0, 20 mM HEPES). In addition, the results of cell imaging of Mg^2+^ in Hl-7701 cells was satisfying.

## 1. Introduction

Many biologically important targets, including metal ions, anions, and amino acids have been successfully detected by fluorescence techniques in vitro and in vivo because of the significant advantages such as the non-destructive characteristic, fast response, and good sensitivity compared with other common detection methods [1,2,3,4,5,6,7]. Among the bioavailable species, Mg^2+^ is the most abundant divalent cation in living cells and plays important roles in many cellular processes, such as DNA synthesis and protein phosphorylation. It also has been involved in many cardiovascular and neurological processes in human body [6,8]. The disorder of concentrations of Mg^2+^ would cause the pathogenesis of many diseases, such as diabetes, hypertension, epilepsy, and Alzheimers [9,10]. So, it is of great importance to detect and monitor Mg^2+^ levels in real time with simple and efficient probes [11,12,13].

Recently, the research on the design and synthesis of Mg^2+^-selective probes has achieved great progress [14,15,16,17]; different receptor groups including crown, calix [4] arene, benzo-chromene, and other functional units have been designed in these probes. However, the reported Mg^2+^ probes still have some disadvantages, particularly poor selectivity between Mg^2+^ and Ca^2+^ due to their similar chemical properties. Compared to the achievement of other targets-selective probes, the development of Mg^2+^ fluorescence probes with high selectivity is still an intriguing challenge due to its silent spectroscopic characteristics and poor coordination ability.

Schiff-base compounds are effective to form stable complexes for different metal ions and anions. Based on these outstanding properties, they have been applied widely in the fields of catalysis, medicine, and fluorescent probes [18,19,20]. Beyond this, photo-induced electron transfer (PET) as a signal mechanism in photophysical processes has been broadly used [7,21,22]. Many reported probes based on PET mechanism contain amine and amide moiety, and the key advantage is easy to prepare and construct “turn-on” type probes for the detection of various targets. In marked contrast to “turn-off” probes, “turn-on” probes could clearly decrease the possibility of false positive signals and improve the sensitivity. For example, Jessica, et al., developed a new probe composing of 2-pyridylhydrazone and 7-hydroxycoumarin moieties, which present a selective fluorescence change to Mg^2+^ [23]. Based on a naphthalene group as the fluorogenic unit and a pyrazole carbohydrazide as the binding unit, Anamika et al. reported a probe for simultaneous determination of Mg^2+^ and Zn^2+^ in MeCN–water solution. The reported probe could be applied in human gastric adenocarcinoma cells [24]. Guang et al. presented a Schiff-base fluorescence probe for Zn^2+^ and Bi^3+^ with different action mechanisms based on PET and ring-open of spirolactam of rhodamine, respectively [25].

Inspired by these mentioned reasons, a Schiff-base compound derived from benzoyl hydrazine was designed and successfully characterized as an “off-on” type Mg^2+^-selective fluorescent probe in this work (Scheme 1). The introduction of O and N donor atoms to the structure of compound **P** improved the coordination ability of the proposed probe to Mg^2+^, more importantly, the formation of C=N also enabled the process of PET to take place effectively. The proposed probe **P** was also successfully used for the detection of Mg^2+^ in Hl-7701 cells.

## 2. Materials and Methods

### 2.1. Reagents and Instruments

The reagents used in the experiment process were commercially available and used directly. The metal ions salts employed are NaCl, KCl, CaCl_2_·2H_2_O, MgCl_2_·6H_2_O, CdCl_2_, HgCl_2_, FeCl_3_·6H_2_O, CrCl_3_·6H_2_O, Zn(NO_3_)_2_·6H_2_O, AgNO_3_, CoCl_2_·6H_2_O, MnCl_2_·4H_2_O, CuCl_2_·2H_2_O, NiCl_2_·6H_2_O, and PbCl_2_. The anions salts employed are NaClO, Na_2_SO_4_, NaNO_3_, Na_2_CO_3_, NaCl, NaAc, NaClO_4_, NaBr, KI, NaSCN, and Na_2_HPO_4_, respectively.

UV-vis absorption spectra and fluorescence emission spectra were recorded on Hitachi U-2910 spectrophotometer and Hitachi 4600 spectrofluorimeter at 25 °C with 1 cm quartz cell, respectively. Mass spectra were collected on a Thermo TSQ Quantum Access Agillent 1100 system. Nuclear magnetic resonance spectra were performed with a Bruker AV 400 instrument (400 MHz), and chemical shifts were given in ppm by using tetramethylsilane (TMS) as internal standards.

### 2.2. Synthesis of the Proposed Probe **P**

**P** was synthesized according to reported methods [26]. 2-aminobenzoichydrazide (1.0 mmol) and 2-hydroxy-1-naphthaldehyde (1.0 mmol) were stirred in ethanol (30 mL) under reflux for 4 h, and then cooled to room temperature. The mixture was filtered off and the yellow precipitate so obtained was used directly. Yields: 87.6%. MS (ES+) *m*/*z*: 306.21 [M + 1]^+^, 328.26 [M + Na]^+^. IR (cm^−1^): 3402.75 (-OH, -NH), 1624.98 (C=O). ^1^H-NMR (*δ* ppm, DMSO-*d*_6_): 12.97 (s, 1H), 11.99 (s, 1H), 9.48 (s, 1H), 8.18 (d, 1H, *J* = 8.56), 7.92 (t, 2H, *J* = 9.30), 7.67 (d, 1H, *J* = 7.80), 7.62 (t, 1H, *J* = 7.64), 7.42 (t, 1H, *J* = 7.42), 7.26 (t, 2H, *J* = 8.46), 6.81 (d, 1H, *J* = 8.24), 6.65 (d, 1H, *J* = 7.56), 6.62 (b, 2H). ^13^C-NMR (*δ* ppm, DMSO-*d*_6_): 165.57, 158.77, 151.36, 146.79, 133.61, 133.37, 132.53, 129.90, 129.01, 128.71, 128.61, 124.42, 121.35, 119.87, 117.58, 115.60, 113.11, and 109.52. (Appendix A).

### 2.3. General Spectroscopic Methods

One-millimeter stock solutions were obtained by dissolving metal salts and **P** with deionized water and dimethyl sulfoxide (DMSO), respectively. The testing solutions were freshly prepared using the stock solutions. For all fluorescent measurements, the slit widths of emission and excitation were both 5 nm; excitation wavelength was set as 415 nm.

### 2.4. Cell Imaging

Hl-7701 cells grown in 6-well coverslips were washed with phosphate-buffered saline (PBS). Mg^2+^ (1 μM) (in PBS) were added to the cells and incubated for 30 min at 37 °C, and then washed with PBS three times followed by the addition of probe **P** (10 μM) and incubation for 30 min at 37 °C. Fluorescence imaging in Hl-7701 and 4t1 cells was recorded by a fluorescence microscope (Olympus FluoView Fv1000).

## 3. Results and Discussion

### 3.1. pH Effects on **P** and **P**-Mg^2+^ System

In order to investigate a suitable pH value for the sensing of **P** to Mg^2+^, the pH titration experiment was performed firstly (Figure 1). The results showed that the fluorescent change of free probe **P** was not obvious in the range of pH 4–11. However, the addition of Mg^2+^ to the solution of **P** caused a fluorescent enhancement at 482 nm in the range of pH 7.4–11, and the maximum was got at pH 10.0. Therefore, UV-vis and fluorescent studies of whole work were carried out in water–ethanol solution (1:9, *v*:*v*, pH 10.0, 20 mM HEPES).

### 3.2. Spectral Study of **P** and **P**-Mg^2+^ System

Good selectivity is a key factor of probes, so the selective experiment of the proposed probe was studied in detail. The fluorescence response of **P** (10 µM) was studied in water–ethanol solution (1:9, *v*:*v*, pH 10.0, 20 mM HEPES) with the addition of tested metal ions Na^+^, K^+^, Mg^2+^, Ca^2+^, Cd^2+^, Co^2+^, Zn^2+^, Pb^2+^, Ni^2+^, Hg^2+^, Cu^2+^, Cr^3+^, Fe^3+^, and Ag^+^ (10 µM) (Figure 2a). The results indicated that only the addition of Mg^2^^+^ generated a significant “turn-on” fluorescence response at 482 nm with a fluorescent intensity enhancement up to 51-fold. It suggested that **P** had better selectivity to Mg^2^^+^ than to other tested metal ions. Meanwhile, no significant variation in fluorescence intensity was found by comparison with Mg^2+^ solution, which had added the same amount of other metal ions and anions (Appendix A). It was gratifying to note that all the tested metal ions and anions have no interference.

Fluorescence titration experiment showed that the fluorescence intensity of **P** (10 μM) increased regularly as the concentration of Mg^2+^ from low to high in water–ethanol solution (1:9, *v*:*v*, pH 10.0, 20 mM HEPES), and the fluorescence intensity of **P** was proportionate to the concentration of Mg^2+^ in the range of 0.9–4.0 µM with a detection limit of 0.3 µM Mg^2+^ (Figure 2b), which indicated that the proposed probe **P** could be used for the detection of environmentally relevant levels of Mg^2^^+^.

The UV-vis spectra of **P** (10 µM) were carried out to further examine the coordination process of Mg^2^^+^ and **P** in water–ethanol solution (1:9, *v*:*v*, pH 10.0, 20 mM HEPES). The addition of Mg^2+^ to the solution of **P** induced an apparent red-shift of absorbance in the UV-vis region and a new peak at 430 nm appeared (Figure 3a), and there is a regular change in the UV-vis spectra at 375 and 430 nm following the introduction of various concentrations of Mg^2+^ to the solution of **P** in water–ethanol solution (1:9, *v*:*v*, pH 10.0, 20 mM HEPES). (Figure 3b). The study clearly suggested that it formed a new complex between **P** and Mg^2+^.

### 3.3. Proposed Reaction Mechanism of **P** with Mg^2+^

In order to study the reaction mechanism and the binding mode of **P** with Mg^2+^, Job’s plot experiment was carried out, and the study revealed that a **P**-Mg^2^^+^ complex was formed in 1:1 molar ratio (Figure 4), and the association constant *K* was determined to be 2.6 × 10^5^ M^−1^ based on the Benesi–Hildebrand method (Appendix A) [27,28,29]. The 1:1 stoichiometry was also supported by the MS spectra of **P**-Mg^2+^ complex, MS (ES+) *m*/*z*: 328.0 assigned to [**P** + Mg^2+^ − H^+^]^+^, 346.1 assigned to [**P** + Mg^2+^ + OH^−^]^−^ and 374.3 assigned to [**P** + Mg^2+^ + CH_3_CH_2_OH − H^+^]^−^ (Appendix A).

To investigate the binding mode, ^1^H-NMR titration was also conducted (Appendix A). Upon addition of Mg^2+^ to the solution of **P**, the signal of phenolic H_a_ at 6.60 ppm disappeared, and H_b_, H_c_, and H_d_ signals broadened after the coordination with Mg^2+^ accompanied by observable chemical shift changes from 12.93 ppm, 9.47 ppm, and 7.65 ppm to 13.08 ppm, 9.76 ppm, and 7.82 ppm, respectively. The shifts of protons of benzene and naphthalene were also marked out in Figure 5. These changes indicated the involvement of the phenolic hydroxyl group (−OH), the nitrogen atom of the imine group (–N=CH), and the O atom of the carbonyl group (–C=O) in the binding of **P** with Mg^2+^. IR spectra of **P**-Mg^2+^ complex was also conducted to study the formation of **P**-Mg^2+^ complex (Appendix A). In the spectra of **P** (Appendix A), characteristic peaks of –OH and –NH appeared at 3402.75 cm^−1^, and that of –C=O was at 1624.98 cm^−1^. However, when probe **P** was completely coordinated with Mg^2+^, −OH peak disappeared, and the stretch vibration of –NH at 3473.03 and 3362.88 cm^−1^ emerged in the IR spectra of **P**-Mg^2+^ complex, and the peak at 1578.36 cm^−1^ belonged to C=N group, and the binding of **P** with Mg^2+^ caused the blue shift of C=O from 1624.98 to 1620.15 cm^−1^ in IR spectra of **P** and **P**-Mg^2+^. Thus, the IR data also supported that –C=O, −OH, and −C=N groups participated in the formation of **P**-Mg^2+^ complex. Thus, based on the obtained results and the reported works [7,18,19], the proposed probe **P** was most likely to coordinate with Mg^2+^ in the mode as shown in Figure 5. The binding with Mg^2+^ blocked the photo-induced electron transfer (PET) mechanism and caused the fluorescence enhancement of **P**.

### 3.4. Preliminary Analytical Application

To further explore the biological applicability of probe **P**, a cell imaging experiment was performed to detect Mg^2+^ in living Hl-7701 cells, and the fluorescence images were recorded by using confocal fluorescence microscopy on an Olympus FluoView Fv1000 laser scanning microscope (Figure 6). The cells were supplemented with only **P** (10 μM) in the growth medium for 30 min, which led to a very weak fluorescence (Figure 6a), suggesting that autofluorescence from the cells could be avoided and very faint fluorescence signal was detected in cells when treated only with **P**. In contrast, when loaded with Mg^2+^ (1 μM), a significant fluorescence change was recorded after the addition of **P** (10 μM) to the cells (Figure 6c), demonstrating that **P** could penetrate cell-membrane and further complex with Mg^2+^ inside the cells, which also clearly illustrated that the interaction of Mg^2+^ with **P** resulted into the “off-on” fluorescence change. Moreover, the bright field images of cell shapes indicated that **P** had low toxicity and good biocompatibility for bioanalysis (Figure 6b). Hence, these results indicated that probe **P** may be utilized as an efficient candidate for in vitro imaging of Mg^2+^ in living cells and potentially in vivo. Meanwhile, to evaluate cytotoxicity of the proposed fluorescent probe **P**, an MTT assay on Hl-7701 cells was performed with the cultured **P** concentrations from 0–10 μM. The cellular viability was ca. 95% in 48 h when treated with 10 µM of **P** (Appendix A), which demonstrated low toxicity to cultured cells.

The subcellular localization of Mg^2+^ in Hl-7701 cells was observed with probe **P** using confocal fluorescence microscopy. The Hl-7701 cells were co-incubated with **P** (10 μM) and Hoechst 33,342 (1 μg·mL^−1^) for 30 min in the same conditions as those used in the cell imaging experiment. It could be concluded that **P** located primarily in the cytoplasm of these living Hl-7701 cells, as represented in Figure 6f.

Furthermore, the cytoplasm location of **P** in living 4t1 cells was also studied by conducting co-localization experiment with mitochondria-specific dye Mito Tracker Red (Figure 7). As seen, the image of the probe **P** (blue panel, Figure 7a) and dye Mito Tracker Red (red panel, Figure 7b) could not merge completely in mitochondria (Figure 7c), so probe **P** localization in mitochondria was excluded (Figure 7c).

## 4. Conclusions

In summary, a simple structure probe derived from benzene hydrazine was characterized as a Mg^2+^-selective probe. The proposed probe showed an “off-on” response in the presence of Mg^2+^ in water–ethanol solution (1:9, *v*:*v*, pH 10.0, 20 mM HEPES). We believe that this study will significantly promote the development of effective Mg^2+^-selective probes for both studies on the effects of Mg^2+^ in environmental or in biological systems.

## Data Availability

Data supporting reported results are available online.

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
