# Peer review of "Synthesis and Characterization of a Mg^2+^-Selective Probe Based on Benzoyl Hydrazine Derivative and Its Application in Cell Imaging"

_molecules, 2021, doi:10.3390/molecules26092457_

Round 1

Reviewer 1 Report

This manuscript reports a new and simple bnzoyl hdrazine drivate P for Mg2+-selctive fluorescent probe. The results are interesting, but intensive improvement is necessary.

  1. To prove the proposed sensing mechanism, various tests such as IR, 1H NMR titration and X-ray are necessary.
  2. I think that the proposed reaction mechanism is not plausible, based on MS spectrum. The MS data showed that the proton of -OH group was deprotonated, suggesting the coordination of the -OH to Mg2+.
  3. Inhibition test should be carried out in the presence of other metal ions.
  4. There are many English mistakes.

Author Response

The authors are very grateful to the reviewer for the valuable comments. The manuscript has been carefully revised in light of the comments. And the full-text, Schemes and Figures are all carefully checked and corrected for improvement. The revised sections have been indicated in red words in the revised manuscript.

Reviewer 2 Report

The manuscript titled "Synthesis and Characterization of a Mg2+-Selective Probe Based on Benzoyl Hydrazine Derivative and Its Application in Cell Imaging" reports the synthesis of (E)-2-amino-N'-(2-(2-hydroxynaphthalen-1-yl)ethylidene)benzohydrazide (compound P) and its characterization as an “off-on” type Mg2+-selective fluorescent probe. The main problem of this manuscript is that the synthesized probe is not a new chemical compound (for example see compound E-3e in J. Med. Chem. 2018 February 08; 61(3): 666–680. doi:10.1021/acs.jmedchem.7b00530). The authors suggest that the main application of this compound may be its use as a fluorescence sensor for in vitro imaging of Mg2+ in living cells. However, the results obtained are not very clear to me. It is known that mammalian cells maintain free cytosolic Mg2+ levels within the range of 0.25-1 mM. If so, NL-7711 cells should also contain approximately the same amount of free magnesium and should show a fluorescent response when treated with compound P. However, fluorescence is absent in Fig. 5a, but intense fluorescence appears for cells pretreated with magnesium salt with a concentration of only 0.001 mM (Fig. 5c), which is obviously insufficient to noticeably change the free cytosolic Mg2+ level in the cell. This concern must be clarified before the manuscript can be accepted for publication. For this reason, I cannot recommend publishing this manuscript in Molecules in its current form.

Author Response

(The authors gave the same response as above.)

Reviewer 3 Report

The authors synthesized a novel fluorescent probe (denoted as P in the paper) for the detection of Mg2+ ions in biological samples. The probe is cell permeable and is selective to Mg2+ over other metal ions. The paper requires revision at a number of points, but overall, it presents sufficient novelty for a publication in Molecules.

Comments

-  (Introduction, 1st paragraph) I suggest to present some examples for cellular processes where Mg2+ plays a key role and some examples for the deseases connected to the disorder of Mg2+ concentration

- (Introduction, last paragraph) If there were fluorescent probes reported in the literature, with structures similar to P which inspired the present work, they should be mentioned

- (Introduction, last paragraph) - What functional groups the authors think to serve as electron donor and acceptor in the PET sensor. The direction of PET should be shown in the structural formula of P in Fig. 2.

- I have not found what was the anion of the Mg salt and the salts of other metal ions used as reference

- It would be more correct to use ‘Fluorescence intensity [a.u.]’ as legend on the y axes of the spectra, instead of ‘Intensity’. Abbreviated forms like ‘I (with subscript) F [a.u.]’ may also be acceptable

- I suggest to calculate the binding constant of the Mg-P complex by a least square fitting of the absorption or fluorescence spectra

- (Fig. 2a) A comparison of the responses to various metal ions could be visualized better in a column diagram like in Ref. 22

- A scale would be helpful on the microscopic images in Fig. 5

Author Response

(The authors gave the same response as above.)

Round 2

Reviewer 1 Report

The authors carried out some tests such as IR and 1H NMR titration to prove the proposed sensing mechanism, but their analyses are not clear. The assignment of NMR peaks are not clear, and IR spectra should be taken and compared for the probe and probe-Mg2+. Therefore, I recommend that the authors would resubmit after intensive improvement of the manuscript is made.

Author Response

Thanks a lot for the valuable comments, and the analysis of 1H-NMR and IR of P and P-Mg2+ complex were intensively studied, which were marked in red in the main text as shown in the attachment file.

Reviewer 2 Report

The authors improved the manuscript and responded to the comments of the reviewers.

Author Response

Thanks a lot for the valuable comments. Intensive improvement of the manuscript has been made according to the comments of the reviewer.